# Continuous Neurophysiologic Data Accurately Predict Mood and Energy in the Elderly

**DOI:** 10.3390/brainsci12091240

**Published:** 2022-09-14

**Authors:** Sean H. Merritt, Michael Krouse, Rana S. Alogaily, Paul J. Zak

**Affiliations:** Center for Neuroeconomics Studies, Claremont Graduate University, Claremont, CA 91711, USA

**Keywords:** attachment, social connection, stress, quality of life

## Abstract

The elderly have an elevated risk of clinical depression because of isolation from family and friends and a reticence to report their emotional states. The present study explored whether data from a commercial neuroscience platform could predict low mood and low energy in members of a retirement community. Neurophysiologic data were collected continuously for three weeks at 1Hz and averaged into hourly and daily measures, while mood and energy were captured with self-reports. Two neurophysiologic measures averaged over a day predicted low mood and low energy with 68% and 75% accuracy. Principal components analysis showed that neurologic variables were statistically associated with mood and energy two days in advance. Applying machine learning to hourly data classified low mood and low energy with 99% and 98% accuracy. Two-day lagged hourly neurophysiologic data predicted low mood and low energy with 98% and 96% accuracy. This study demonstrates that continuous measurement of neurophysiologic variables may be an effective way to reduce the incidence of mood disorders in vulnerable people by identifying when interventions are needed.

## 1. Introduction

Depression is a primary public health concern globally. The elderly are particularly vulnerable to depression due to age-related neural atrophy, hypertension, and social isolation [1,2,3]. There are a variety of ways to inhibit the onset of depression, including social support, psychological counseling and pharmacotherapy [4,5,6,7]. Such interventions are more effective if a decline in mood can be identified before a major depressive episode occurs [8]. The ability to passively assess mood states using technology would be an important public health advance [9]. Several approaches to predicting low moods are now being investigated. For example, using smartwatches and machine learning to analyze sleep as depressive episodes are associated with disordered sleep patterns [10].

Chronic low mood increases morbidity and mortality, especially in older adults [11,12]. When people experience low moods that last for two weeks or more, they are diagnosed as clinically depressed [13]. The lifetime incidence of depression is 14.6% for adults in developed countries [14], and women are approximately twice as likely as men to have an episode of depression [15]. In the U.S., those aged 65 and older have a one in four depression risk [16,17]. Life events can increase the likelihood of depression in seniors, including declining health, financial straits, losses of loved ones, reduced social interactions, inadequate healthcare, and the inability to participate in activities [18,19,20,21,22]. Depression in old age is also a risk factor for dementia [23].

On the other hand, positive affect has a host of favorable impacts on health in the elderly, including a lower risk of cardiovascular disease [24], a reduction in pain [25,26], increased exercise [27], improved immune function [24,28], and better social relationships [29]. It is likely that the causal flow connecting positive mood to improved health is bidirectional [30,31] and depends in part on one’s genetics [32]. The importance of mood states on healthspan demands a more fundamental understanding of the causes of mood variations [33]. As this research advances, data that quantify activities and physiology using wearable technologies has exploded. These new data may make it possible to predict mood states in the elderly using physiology in order to create interventions to reduce or eliminate the degradation of health from persistent negative affect.

Not only is there a great need to predict moods, but the use of neural data obviates the need to constantly query individuals. Self-reports tend to be inaccurate, especially in the elderly [34,35]. Colloquially, people may be “worried to death,” and indeed, there is an extensive literature relating negative mood states and clinical depression to anxiety [36,37]. Anxious individuals have elevated activity of the sympathetic autonomic nervous system [38]. Typical measures of phasic sympathetic tone include heart rate and electrodermal activity. Most pharmacotherapies for depression reduce sympathetic tone, along with other effects [39]. While basal sympathetic tone varies substantially across individuals [40], measures of sympathetic nervous system activity are a reliable prodrome for depression [41,42,43].

When individuals, including the elderly, are anxious, their ability to enjoy life is inhibited [44]. Indeed, social activities in the elderly reduce anxiety [45,46,47,48] as do supportive relationships [5,49,50]. Seniors can create opportunities for social interactions by volunteering [51], investing in friendships [5], and joining group activities [50]. Socially active seniors are less likely to suffer from anxiety [52].

Depressive symptoms in seniors may arise when individuals no longer enjoy activities (anhedonia). However, even with observation, it may take weeks or months to correctly classify an individual as depressed since variations in moods are common. When depressive symptoms are identified early, the prognosis for patients is substantially improved [8]. The interaction between the quality of social activities and mood has the potential to be measured neurophysiologically [53,54].

Low mood and depression have been predicted by applying artificial intelligence techniques to digital traces [55]. A typical approach employs natural language processing for social media posts and chats [56,57]. Other research has used machine learning to predict moods using song choices [58], street views [59], pictures of faces [60], and images from video conferences [61]. These approaches are convenient because they use publicly available data without the need for direct measurement of participants. However, a shortcoming of this approach is the use of surveys of nonparticipants to assess the moods of participants. This induces bias in the dependent variables as people inconsistently identify others’ moods [62,63]. As a result, the predictive accuracy of these models seldom exceeds the 70–80% range.

The present exploratory study sought to relate self-reported moods to neurophysiologic data collected directly from study participants. This is a difficult task as consciously-filtered self-report measures are typically unrelated to neural activity [35]. Mood was assessed using retrospective daily self-reports, while neural measures were obtained at 1Hz for 8–10 h per day. The first step in creating a potential early detection measure for melancholia is to determine if neurophysiologic measures are associated with mood. The present study used a sample of healthy seniors, rather than a clinical population, in order to test the hypothesis that a combination of neural measures derived from a wearable sensor would predict changes in mood states.

## 2. Materials and Methods

*Participants.* Twenty-four participants were recruited from a Texas residential living facility. A flyer was circulated requesting signups for the study after obtaining permission from facility management. Those with serious health conditions were excluded from participation. Residents were provided with Apple Watch 6s loaded an app called Immersion Mobile to collect neurophysiologic data. Data were collected for 20 days between 18 January and 24 February 2021 for up to 10 h each day. All participants who started the study completed it. The initial analysis averaged neurophysiologic data for each day, resulting in 480 observations. Next, we generated average hourly data producing 2478 observations to explore whether higher frequency data improved predictive accuracy.

*Procedure.* Participants were sent an email every day at 6 a.m. and asked to complete an online survey reporting their mood, health, and energy the day before. If no response was collected by noon, participants were reminded via email and text to complete the survey. Missing data from the self-reports was <4%. Retrospective self-reports were used to capture perceptions of the previous day’s mood because contemporaneous self-reports may induce a bias towards one’s acute mood state, but can reduce accuracy due to poor recall and misattribution of arousal [64,65]. This approach decreases the likelihood of significant associations to physiologic signals associated with retrospective mood reports. In order to reduce the burden of data collection, which required wearing a smartwatch daily and charging it overnight, the only additional information obtained from residents was their biological sex and whether they were ill.

*Neurophysiology.* A commercial platform (Immersion Neuroscience, Henderson, NV, USA) was used to measure neurophysiologic responses collected at 1Hz. The independent variables obtained from the Immersion platform were average Immersion for each day and average psychological safety. Neurologic Immersion combines signals associated with attention and emotional resonance and measures the value the brain places on social experiences [66,67] The attentional response is associated with dopamine binding to the prefrontal cortex, while emotional resonance is related to oxytocin release from the brainstem [67,68,69]. The Immersion Neuroscience platform infers neural states from the activity of the cranial nerves using the downstream effects of dopamine and oxytocin on cardiac rhythms [66,70]. The data are sent to the cloud continuously, and the Immersion platform provides an output file used in the analysis. We chose to measure neurologic Immersion for this study because of the relationship between the quality of social connections and mood in the elderly [31].

A second neurologic measure from the Immersion platform, psychological safety (PS), measures parasympathetic tone, which reflects stress levels that impact mood [71]. In addition, we created a variable called peak Immersion, defined as
$$\int_{t=0}^{T}\;(v_{it}>M_{i})d_{t}/Im_{i}$$
where *v_it_* is the hourly average neurophysiologic Immersion for each participant in day *i* at time *t* to the end of the day at time *T*, *M_i_* is the median of the average hourly time series of Immersion for day *i* plus the standard deviation of the hourly data for day *i* for each participant, and this is divided by the sum of total Immersion *Im_i_* for each person for each day *i*. That is, peak Immersion (PI) cumulates the highest Immersion moments for an individual during the day capturing high-value social experiences relative to the Immersion from total social experiences.

*Self-Report Measures*. Mood was assessed by averaging four questions on a 1–5 scale (cheerful, stressed, lonely, energy) with stressed and lonely reverse coded using an abbreviated version of the Profile of Mood States (POMS) [72,73]. Mood was defined as “Low” if it was lower than the median value 4 and was labeled “High” for values greater than or equal to 4. Due to the moderate period of data collection, we did not seek to identify clinical depression, a state in which people remain for a period of time. Rather, we sought to identify variations in mood troughs that may be a prodrome for clinical depression. Mood has only moderate interpersonal and intrapersonal variation (Intrapersonal CV: 10.80%, Interpersonal CV: 16.26%). Energy was used as a second dependent variable because social activities are typically energizing. Energy has more variation than Mood (Intrapersonal CV: 23.64%, Interpersonal CV: 31.45%). “Low” energy was defined as a value of the median of 3 or lower and “High” was for values greater than the median. The only demographic data collected in this exploratory study was biological sex. A binary variable identifying if the participant was sick (Sick) was collected as a control.

*Statistical Analysis*. Multiple techniques were used to extract as many insights from the data as possible and to establish the robustness of our findings. While the data constitute a panel, both Mood and Energy show little time-series variation. Statistical tests (Results) indicate that each observation can be analyzed as an independent observation. The analysis begins with t-tests and correlations relating Mood and Energy to neurologic variables averaged for each day. We tested mean-corrected differences in neurologic variables for low and high Mood and Energy. An ordinary least squares (OLS) regression was estimated to predict participants’ Moods and Energy using Immersion, psychological safety, and peak Immersion as independent variables and including Sick as a control. Logistic (logit) regressions were also estimated to establish predictive accuracy. In addition, since neural variables are expected to be moderately correlated, principal components analysis (PCA) was used to seek to improve predictive accuracy. We used nine input features for the PCA: immersion, peak immersion, psychological safety with one- and two-day lags. These analyses were performed using the Stata 17 software package (StataCorp, College Station, TX, USA).

Average hourly Immersion and psychological safety were used to build machine learning (ML) models to assess predictive accuracy compared to models using daily data. The ML models included regularized logistic regressions, random forests (RF), and support vector machines (SVM). Participants missing more than four hourly observations were dropped from the analyses for that day. In order to avoid further data loss, separate models were trained using Immersion and psychological safety. The *tsfresh* package in Python was used to extract features from the hourly time-series data. The *tsfresh* package extracts up to 794 time-series features and tests these for relationships to the dependent variables [74]. Features that had bivariate correlations with the dependent variables that had ps > 0.05 were discarded.

The unbalanced panel led us to use the synthetic minority oversampling technique (SMOTE) to balance the ratio of high and low values for mood and energy [75]. We then split the SMOTE data into a training (75% of the complete dataset) and testing (25%) set. The former set was used to tune hyperparameters using a 5-fold cross-validation grid search (the *GridSearchCV* function in the *sklearn* package). Table A1 in Appendix A shows the parameters tuned for the machine learning models.

We then estimated the models on the test set and the pre-SMOTE data. The area under the curve (AUC) was assessed for model fit, and true positive and true negative rates (Precision and Recall, respectively) and their average was used to measure accuracy. Possible overfitting was tested using 5-fold cross-validation on the entire data set. Accuracy and fit were examined for contemporaneous data and for a two-day lag prior to the self-report to determine the consistency of the neural measures.

## 3. Results

*Descriptive Statistics.* Mood and Energy had means on the high end of the reporting range and moderate spreads (Mood: M = 3.84 SD = 0.65, Figure 1; Energy: M = 3.04 SD = 0.97, Figure 2). The neurophysiologic variables fell into expected ranges (Immersion: M = 36.29, SD = 3.51, Range = [28.85, 48.91]; PI: M = 0.0076, SD = 0.0079, Range = [0, 0.054]; PS: M = 18.14, SD = 5.51, Range = [8.80, 40.70]).

*Time Series Aspects*. We tested if the time series of the two dependent variables contained information that should be included in our analyses. If participants’ time series were random walks, then the time series component of the analysis could be ignored. We used a Fisher-type unit-root examination using augmented Dickey–Fuller tests in which the null hypothesis is that the time series has a unit root [76]. The estimation indicated that both the one- and two-day lagged time component of the dependent variables did not affect the current value of the dependent variables (Table A2). As a result, each observation can be analyzed independently, ignoring the time dimension.

*Mood*. Among the three physiologic variables, only PI was statistically related to Mood (Immersion: r = 0.091, *t* (318) = 1.25, *p* = 0.105; PI: r = 0.16, *t* (318) = 2.66, *p* = 0.004; PS: r = −0.001, *t* (318) = −2.23, *p* = 0.990). Participants with high Moods had higher PI (M_high_ = 0.009, M_low_ = 0.007; *t* (318) = −2.01, *p* = 0.02) and no difference in the other physiologic variables (Immersion: M_high_ = 36.14, M_low_ = 36.35; *t* (318) = 0.4812, *p* = 0.68; PS: M_high_ = 1.77, M_low_ = 1.82; *t* (318) = 0.721, *p* = 0.7644). Moods in males averaged 5.5% higher than that of females (M_male_ = 4.0, M_female_ = 3.79; *t* (318) = −2.384, *p* = 0.008).

An ordinary least squares regression was used to test if Immersion, PS, and PI were related to participants’ Moods while controlling for sex and being sick. Thirteen values for PI exceeded 5 SD above the mean and these observations were removed. Due to the high variance of PI, it was standardized by subtracting the mean and dividing by the sample standard deviation using Stata’s *standardize* function. The regression was statistically significant (R^2^ = 0.173, F (5, 298) = 12.54, *p* =.000) and showed that Immersion and PI were positively associated with Mood, but PS was not (Immersion: ß = 0.403, *p* = 0.002; PI: ß = 0.155, *p* = 0.003; PS: ß = −0.091, *p* = 0.264). Variance inflation factors (VIFs) were within acceptable limits showing the estimation was well-specified.

A logistic regression was then estimated to examine the predictive accuracy of the physiologic variables. As above, the logistic regression showed that Immersion and PI were significant along with moderately large odds ratios (OR) (Immersion: OR = 3.29, *p* = 0.014; PI: OR = 1.533, *p* = 0.026; PS: OR = 1.136, *p* = 0.671; Table 1) with predictive accuracy of 67.76%.

We used the logistic regression estimates to examine when a participant was likely to have low Mood. For a woman who is not sick, there is a 75% likelihood she has low Mood when Immersion is 26.50 or lower, holding PS and PI at their medians. A man has a 75% chance of having low Mood when his Immersion is 12.36 or below for the median values of PS and PI. The threshold values of Immersion producing a 75% chance of low Mood are higher when PI and PS are lower (Table A3).

*Principal Components*. The first principal component (PC1) was loaded primarily on contemporaneous values and 1- and 2-day lags of PS and Immersion. The second principal component (PC2) was loaded on contemporaneous values and 1- and 2- day lags of PI. The third principal component (PC3) was loaded almost entirely on Immersion with a one-day lag (Table A4). Regression analysis was used to test the relationship between the first three principal components and the dependent variables. All three PCs were statistically significant (PC1: ß = 0.0599, *p* = 0.002; PC2: ß = 0.142, *p* = 0.000; PC3: ß = -0.095, *p* = 0.028). In addition, all three principal components were significant predictors of high and low Mood in a logistic regression (PC1: OR = 1.375, *p* = 0.000; PC2: OR = 1.963, *p* = 0.000; PC3: OR = 0.642, *p* = 0.026). The use of principal components improved the predictive accuracy of Mood to 74.81% (Table A5).

*Energy*. The second dependent variable, Energy, was centered on the mean (M = 3.038, SD = 0.966; Figure 2). Immersion, PI and PS were all related to Energy bilaterally (Immersion: r = 0.184, *t* (318) = 2.63, *p* = 0.009; PI: r = 0.233, *t* (318) = 4.84, *p* = 0.000; PS: r = 0.151, *t* (318) = 2.72, *p* = 0.0068). Immersion was 5.3% higher in those with high Energy compared to Immersion in low Energy participants (M_high_ = 3.76, M_low_ = 3.57; *t* (318) = −4.138, *p* = 0.000). PI and PS were also higher in those with high Energy compared to low Energy participants (PI: M_high_ = 0.01, M_low_ = 0.006; *t* (318) = −3.925, *p* = 0.000; PS: M_high_ = 1.99, M_low_ = 1.74; *t* (318) = −3.7122, *p* = 0.000). There were no sex differences in participants’ Energy (M_male_ = 3.08, M_female_ = 3.02; *t* (318) = -0.442, *p* = 0.659).

Regression estimates revealed significant associations for Immersion and PI with Energy (Immersion: ß = 0.498, *p* =.010; PI: ß = 0.268, *p* = 0.001; PS: ß = 0.122, *p* = 0.324). A logistic regression for high and low Energy found Immersion and PI were significant predictors (Immersion: OR = 3.11, *p* = 0.020; PI: OR = 1.655, *p* = 0.011; PS: OR = 1.66, *p* = 0.094; Table 2) and had predictive accuracy of 74.67%.

The logistic regression estimates were used to identify predictors for high and low Energy. For a woman who is not sick, there is a 75% or greater likelihood she has low Energy when Immersion is 35.80 or below, setting PI and PS to their medians. The corresponding threshold for men is an Immersion value of 40.17 (Table A6).

Next, we used ordinary least squares to test if PC1, PC2, and PC3 were related to participants’ Energy. PC1 and PC2 were both associated with Energy, though PC3 was not (PC1: ß = 0.143, *p* = 0.000; PC2: ß = 0.266, *p* = 0.000; PC3: ß = −0.116, *p* = 0.076). A logistic regression for high and low Energy found a significant association for PC1 and PC2 as in the OLS results (PC1: OR = 1.476, *p* = 0.000; PC 2: OR = 2.26, *p* = 0.000; PC3: OR = 0.889, *p* = 0.485) and produced a predictive accuracy of 74.81% (Table A7).

*Machine Learning Models*. We first estimated models using hourly data (2478 observations) to predict Mood. Extracting significant components produced 18 features from Immersion and five from PS. Then, we balanced the data with SMOTE and tuned the hyperparameters. The models were estimated on the training set with the following parametric restrictions for both Immersion and PS: logit (C = 100, penalty = “l2”), RF (max_features = ‘log2′, ‘min_samples_leaf’ = 2, ‘min_samples_split’ =2), and SVM (C = 100, kernel = ‘rbf’).

The ML estimations classifying Mood fit the data well in the test set (AUCs > 0.90). The RF algorithm produced the best fit for the test data set (AUC = 1.00), whereas the fit for logit (AUC = 0.71) and SVM (AUC = 0.82) declined. Using PS as the explanatory variable for the ML estimations had similarly high goodness-of-fit on the test set for RF (AUC = 0.90) and SVM (AUC = 0.94), but only moderate fit for regularized logit (AUC = 0.76). AUC fell for all three ML models when classifying the test observations (Logit: 0.59, RF: 0.78, SVM: 0.81). To account for possible overfitting, we used 5-fold cross-validation on the full SMOTE data. All models maintained high scores across the five folds indicating they were not overfit. Predictive accuracy for the ML models using Immersion was very high for the observed data; correct classification of Mood ranged from 99 to 100%. Using PS in the ML estimation was nearly as accurate, ranging from 98 to 100% (Table A8).

Repeating the ML estimations using Energy as the dependent variable, 27 features from the Immersion data and 11 from PS were extracted using logit (C = 100, penalty = “l2”), RF (max_features = ‘sqrt’, ‘min_samples_leaf’ = 2, ‘min_samples_split’ = [2, 5]), and SVM (C = 10, kernel = ‘rbf’). Immersion explained the Energy data moderately well, with RF performing the best on the test (AUC = 0.80) and observed set (AUC = 0.93) and SVM and logit performing adequately (SVM: 0.78; 0.86; Logit: 0.73; 0.78). Cross-validation shows that all the models perform moderately well but with high variation (SDs > 0.044).

Predictive accuracy for the Immersion ML models of Energy using the observed data was high, though below the classification accuracy of Mood, with the RF being most accurate (logit: 82%; RF: 95%; SVM: 90%). The RF and SVM models of Energy using PS as the independent variable had similar accuracy to the Immersion models, while the regularized logit was fairly inaccurate (logit: 65%; RF: 94%; SVM: 88%, Table A9).

*Two Day Lag Machine Learning Models.* We examined the predictive accuracy of Mood with neural data two days prior to the self-report to extend the principal component findings (2422 observations). Seventeen features were extracted from Immersion, and eight features were extracted from PS. Hyperparameters for Immersion were tuned using: logit (C = 1, penalty = “l2”), RF (max_features = [‘log2′;’sqrt’], ‘min_samples_leaf’ = 2, ‘min_samples_split’ = [5, 2]), and SVM (C = 10, kernel = ‘rbf’).

The two-day lagged goodness of fit mirrored the contemporaneous estimations. Fit was quite good for Mood using Immersion for the test and observation datasets for RF (AUC = 1.00; 0.96) and SVM (AUC = 0.99; 0.96). The logit performed well on the test set, but only moderately on the observed set (AUC = 0.94; 0.74). The fits for the estimations using PS were also good, though they declined somewhat from the test to observed data (RF: AUC = 0.91, 0.79; SVM: AUC = 0.96; 0.88). As with the Immersion estimates, the logit performed poorly (AUC = 0.67; 0.57; Table A10). The cross-validation showed that all models fit well except for logit (AUC = 0.78, SD = 0.055).

Next, we used Immersion and PS to predict Energy with two-day lagged data. This procedure extracted 30 features from Immersion and four from PS. Hyperparameters were: logit (C = [1, 10], penalty = “l2”), RF (max_features = [‘log2′, ’sqrt’], ‘min_samples_leaf’ = 2, ‘min_samples_split’ = 2), and SVM (C = 100, kernel = ‘rbf’). All Immersion and PS models did moderately well on the test dataset (AUCs ≥ 0.84). Indeed, the RF (AUC = 0.93) and SVM (AUC = 0.94) Immersion models fit better on the observed data set than on the training set. The PS models performed moderately well on the test data set, but only RF did so on the observed dataset (AUC = 0.83).

Predictive accuracy continued to be quite high using hourly data lagged two days. Immersion, in particular, strongly predicted Mood using the lagged data for all three ML estimations (logit: 92%, RF: 99%, SVM: 99%). The two-day lagged RF and SVM models were nearly identical in their accuracy compared to the contemporaneous data estimates. Predictive accuracy of Mood using two-day lagged PS was lower compared to the use of Immersion and was similar to the contemporaneous estimates. The RF and SVM models had high accuracy, while the logit was moderately accurate (logit: 76%; RF: 95%; SVM: 98%). The predictions for individuals’ Energy using Immersion were 95% and 96% accuracy (RF, SVM) for the observed data, while the logit model performed only moderately (79%). The two-day lagged predictions for Energy using PS, as with Mood, were less accurate, with the RF model being the best (logit: 57%, RF: 87%, SVM: 68%; Table A11).

## 4. Discussion

The key contribution of the present exploratory study is to demonstrate that high-frequency neurophysiologic measures accurately predict self-reported emotional states. Passive and continuous data collection, as used here, appears to be an effective way to monitor emotional wellness. Our analysis shows that declines in emotional states can be predicted two days in advance with high accuracy. Such data make it possible for family members or clinicians to check in on the elderly in order to halt a decline before a potential mental health crisis occurs. More generally, we have shown that neural measures can be used to monitor the quality of life in seniors and perhaps other vulnerable populations. Neural predictors of emotional states can also be used to identify the physiological processes inhibiting satisfaction with one’s life so that interventions are focused and effective [77].

The results are surprising because 1Hz neurophysiologic data were averaged into hourly and daily measures that we expected would return to a long-term equilibrium. However, this was not the case. Daily observations were statistically independent of each other, yet had predictive value as a group. Estimations using daily data predicted low Mood and low Energy with 68–75% accuracy using just three neurophysiologic variables in a standard logistic regression, controlling for illness and sex. The over-2000 hourly observations led us to estimate ML models for Mood and Energy. These models were extremely accurate, correctly classifying the dependent variables with 84–100% accuracy and were not overfit.

The most predictive measure, Immersion, appears to capture the neural value of social experiences, a key aspect of flourishing [78,79]. Neural measures such as Immersion are necessary because the ability to self-report the quality of relationships is difficult and is nearly impossible with the granularity of the neurophysiologic measures used here. Measurement of social connections is vital because they increase the positive affect [80,81] and improve life satisfaction [82,83,84].

Psychological safety on a daily basis did not influence Mood or Energy. However, this variable accurately predicted both dependent variables when measured hourly. Extensive research has related psychological safety and the absence of anxiety to improved mood [85]. Psychological safety regulates people’s emotional well-being by motivating a desire for social support to reduce anxiety [86,87,88]. When anxiety is reduced, the quality of social relationships improves, enhancing healthspans [89,90].

As expected, sickness reduces Mood and Energy. The desire and ability to socialize are reduced with illness, inhibiting the benefits of socializing [91,92,93]. Sickness negatively affects the quality of life in the elderly in part by inducing negative moods [94]. Chronic illness reduces the independence and mobility of senior citizens [95] affecting their ability to engage with others [91,96]. The physiologic variables in the predictive models of Mood and Energy were significant even when removing the effect of sickness.

The use of a commercial Neuroscience as a Service (NaaS) platform allowed us to exploit fully processed neurophysiologic measures that were captured at scale. This approach makes it easy for other researchers to replicate and extend our findings without having to buy expensive equipment or process high-frequency data. Nevertheless, this study has a number of limitations that should be addressed in subsequent research. The number of participants was small even while the number of observations was high. The sample population studied was fairly homogenous and came from a single retirement home. While we focused on predicting troughs in Mood and Energy in psychologically healthy adults, these states do not necessarily lead to clinical depression [97]. Future research should apply the methodology here to other vulnerable populations, including those with diagnosed mental illnesses, to assess its predictive accuracy in these populations. Lastly, we used two measures from the Immersion platform (Immersion, PS) and derived another measure (PI). Future research could derive additional measures from these data or, as we have done, rely on ML approaches to extract as much predictive value from the variables we used.

## 5. Conclusions

As people age, their moods tend to decline [98,99]. We have shown that troughs in Mood and Energy can be accurately predicted with off-the-shelf wearables and data from a commercial software platform. The present study did not seek to identify clinical depression, which would require longer data collection and more individuals. Rather, we hope that our findings will motivate researchers to identify the causal factors that threaten emotional wellness in older adults and other vulnerable populations. Future research should assess our techniques for those with diagnosed depression who are at risk of recurrence as well as professionals who face chronically high levels of stress, including first responders, clinicians, and members of the military. Accurate measurement is the first step toward improving emotional health so that people can live happier, healthier, and longer lives.

## Figures and Tables

**Figure 1 brainsci-12-01240-f001:**
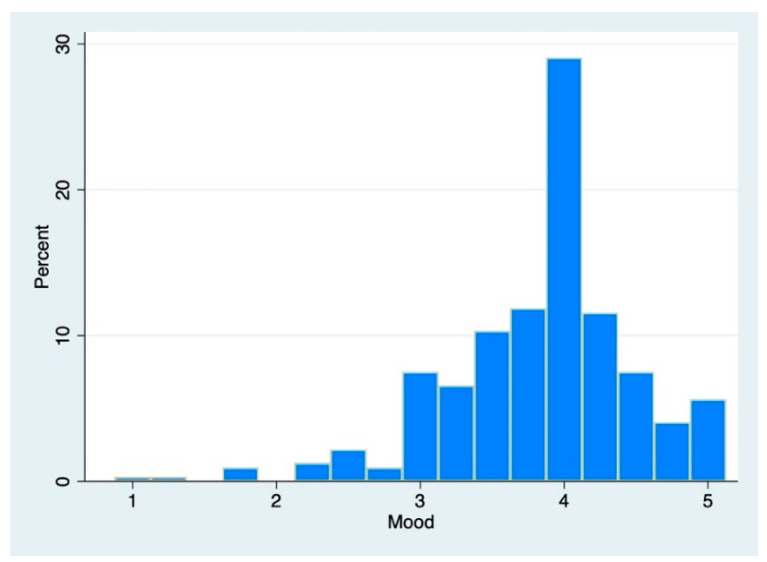
The distribution of participants’ moods.

**Figure 2 brainsci-12-01240-f002:**
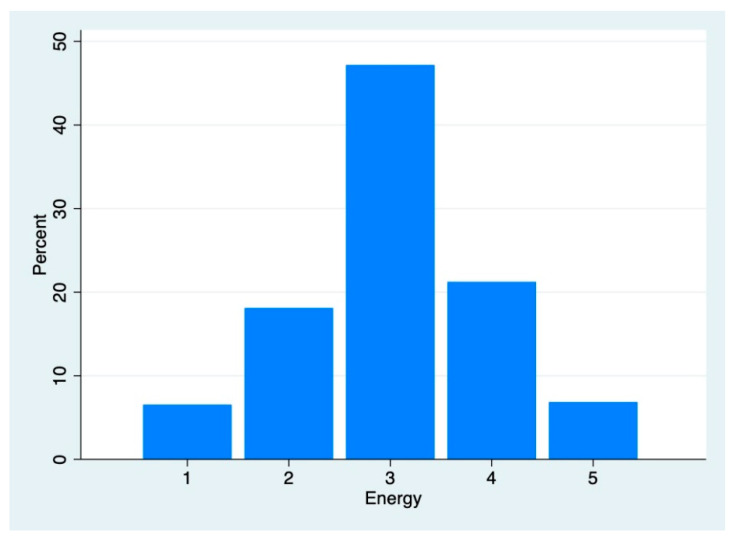
Distribution of participants’ energy.

**Table 1 brainsci-12-01240-t001:** OLS estimation showed that Immersion and peak Immersion have positive effects on Mood. When predicting high and low Mood using a logit estimation, only peak Immersion was significant. Sick and Male are controls. Standard errors are in parenthesis and * = *p* < 0.05, ** = *p* < 0.01, *** = *p* < 0.001.

Variable	*OLS*	*VIF*	Logit	Odds Ratio
Immersion	0.403 * (0.126)	1.60	1.19 (0.484)	3.296
PS	−0.091 (0.081)	1.58	0.127 (0.301)	1.136
PI	19.63 * (6.52)	1.03	0.428 * (0.192)	1.153
Sick	−0.922 *** (0.152)	1.02	−3.889 (1.14)	0.020
Male Intercept	0.285 (0.087) 3.068 * (0.900)	1.09	1.687 (0.370) −0.535 (3.139)	5.403
*F*-value	12.54		Likelihood ratio χ^2^	55.42
*p*-value	0.000		*p*-value	0.000
R-squared	0.174		Pseudo R-squared	0.134

**Table 2 brainsci-12-01240-t002:** Energy is positively associated with Immersion and peak Immersion in an OLS estimation. Sick and Male are controls. A logit estimation for high and low Energy confirms that Immersion and peak Immersion are significant predictors. Standard errors are in parentheses and * = *p* < 0.05, ** = *p* < 0.01, *** = *p* < 0.001.

Variable	*OLS*	*VIF*	Logit	Odds Ratio
Immersion	0.499 * (0.192)	1.60	1.134 * (0.488)	3.11 *
PS	0.122 (0.123)	1.58	0.507 (0.302)	1.66
PI	33.95 ** (9.95)	1.03	0.504 ** (0.199)	1.66
Sick	−0.594 ** (0.232)	1.02	−1.733 (1.06)	0.177
Male	0.159 (0.132)	1.09	−0.504 (0.394)	0.604
Intercept	0.764 * (0.614)		−5.888 ** (1.593)	
*F*-value	7.04		Likelihood ratio χ^2^	37.88
*p*-value	(0.000)		*p*-value	0.000
R-squared	0.106		Pseudo R-squared	0.107

## Data Availability

The data are available on request from the senior author (P.J.Z.).

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
