# Peer review of "Continuous Neurophysiologic Data Accurately Predict Mood and Energy in the Elderly"

_brainsci, 2022, doi:10.3390/brainsci12091240_

Round 1
Reviewer 1 Report
The article's main concept:
This article attempts to propose to predict Mood and Energy states of people using Neurophysiologic Data obtained from an apple watch, as input data for the machine learning (ML) techniques. Neurologic Immersion, psychological safety (PS) and peak Immersion (PI) are the features, which have been used for analyzing the Mood and Energy level of participants. Principal Components Analysis (PCA) has used to remove the possible correlated features, and different ML techniques are used for predicting process.
It is an interesting study, and shows high impact in the relevant fields. However, the article missing several points, which need to be considered.
Overall Comment:
The state-of-the-art needs to be improved. Since the idea is using technology and AI for prediction process, It would be interesting to see the relevant published studies which were used AI approaches.
Section of “Materials and Methods”, is needed to explain more clear the data collection process, feature extraction, AI features like PCA and MLs. For instance, how many features are extracted for prediction process. PCA is a reduction features approach, which map input features to other type of features. Thus, it would be interesting to explain, how much is your input feature size. It would be great to use only table to show numbers and parameters such as different ML with different parameters, which authors explain it, in the text. It’s difficult to follow.
Explain more clear, about the proposed approach to use different prior data for prediction process.
Section “results”
Again, my suggestion to authors is to avoid to use only table to present the numbers of different parameters. Most part of the text mentioned the numbers which part of them are also in the tables. Meanwhile several abbreviations are not defined.
Author Response
- The state-of-the-art needs to be improved. Since the idea is using technology and AI for prediction process, It would be interesting to see the relevant published studies which were used AI approaches.
We have added a paragraph in the Introduction about other research that seeks to predict mood states.
- Section of “Materials and Methods”, is needed to explain more clear the data collection process, feature extraction, AI features like PCA and MLs. For instance, how many features are extracted for prediction process. PCA is a reduction features approach, which map input features to other type of features. Thus, it would be interesting to explain, how much is your input feature size. It would be great to use only table to show numbers and parameters such as different ML with different parameters, which authors explain it, in the text. It’s difficult to follow.
We have added this information to the ms.
- Explain more clear, about the proposed approach to use different prior data for prediction process.
We have added a paragraph in the Introduction about other research that seeks to predict mood states.
Section “results”
- Again, my suggestion to authors is to avoid to use only table to present the numbers of different parameters. Most part of the text mentioned the numbers which part of them are also in the tables. Meanwhile several abbreviations are not defined.
We have ensured all abbreviations are defined.
Tables: We have sought to make the paper readable by including sufficient information in the text that the results can be followed, with additional detailed for interested readers in tables.
Reviewer 2 Report
Review Report
Does the introduction provide sufficient background and include all relevant references?
The Introduction section of the manuscript provides the theoretical background of the research. This area makes a detailed photograph of the study context that synthesizes relevant and recent literature related to the topic.
Is the research design appropriate?
In this manuscript, design research aims to solve a current problem: how to predict mood changes using a sample of healthy elderly rather than a clinical population through a wearable sensor.
Are the methods adequately described?
The section Materials and Methods is described in a complete, detailed, and simple way. The five different subsections make the manuscript more comprehensible to the reader.
Are the results clearly presented?
In the Results section, the analysis results are described in a prominent, concise, and precise manner. In addition, even the figures and tables help significantly understand the results.
Are the conclusions supported by the results?
The manuscript provides essential conclusions and implications in the section “Discussions.”
The conclusions are supported by the results, which are widely discussed quantitatively and qualitatively.
Author Response
Thank you for the feedback.
Reviewer 3 Report
This is an article about a very interesting subject, which is the question if continuous neurophysiologic data can be used as a predictor for mood and energy in the elderly. The manuscript is generally very well structured.
English language and style are fine but there are some minor issues that need to be addressed before publication (e.g. in line 358 the term “to captures” should be replaced by “to capture”).
The introduction describes the background of this study in a comprehensive manner.
“Materials and methods” section is descriptive enough and well-structured too.
The results are very interesting and, to my opinion, well presented.
Perhaps the conclusions should be written in a separate section from discussion, making them clearer and more comprehensible to the readers.
Finally, the authors should include in the paper the necessary authors’ statements such as “Funding”, “Institutional Review Board Statement”, “Informed Consent Statement”, “Data Availability Statement” and “Conflicts of Interest”.
Author Response
- Perhaps the conclusions should be written in a separate section from discussion, making them clearer and more comprehensible to the readers.
We appreciate this suggestion and have added a Conclusion section.
- Finally, the authors should include in the paper the necessary authors’ statements such as “Funding”, “Institutional Review Board Statement”, “Informed Consent Statement”, “Data Availability Statement” and “Conflicts of Interest”.
These have been added to the text.
Reviewer 4 Report
The present work, entitled “Continuous Neurophysiologic Data Accurately Predict Mood and 2 Energy in the Elderly” shows that neurophysiologic measures using a commercial platform predict self-reported emotional states in senior citizens. Moreover, the emotional state can be predicted even two days in advance. In particular, machine learning procedures applied to neurophysiological data resulted in a 99% and 98% of accuracy for predicting low mood and low energy. The authors propose this methodology as a valid tool to detect and reduce the incidence of mood disorders in vulnerable people.
This is an interesting work. The manuscript is well written and scientifically sound. Nevertheless, I have some commentaries for the authors:
Participants:
Please, provide information about the recruitment procedure, as well as inclusion/exclusion criteria. The authors mention that the only demographic data collected was sex. Why other demographic information, particularly participants’ socioeconomic and health status was not collected?
Please, provide information about how the current study accomplished the Declaration of Helsinki, domestic legislation for human research and the university ethics committee approval.
Please provide information about participants abandonment and failure rates to complete the online survey.
The survey items employed to record participants mood, health and energy should be provided or indicate whether it has been published elsewhere. Has such questionnaire been validated? Psychometric properties of the questionnaire should be provided.
Procedure:
Although some references to previous articles are provided, a more detailed description about the construct “Immersion” and its physiological bases should be given. Furthermore, some references are missing in the References section (line 117: Zak & Barraza, 2018) and (line 119: Zak & Nowack, 2021; Zak et al. 2021).
When (daytime) were the physiological measures registered?
Self-Report Measures:
What is the psychometric soundness of the items included in the questionnaire?
How the questionnaire correlates with a diagnostic of depression?
Discussion:
The authors acknowledge that a diagnostic of depression may take months as result of the variability of mood. I understand their model is proposed as a tool to help in the detection and subsequent treatment of depression in elders. Nevertheless, they have used a sample of healthy seniors. As a limitation of the study it should include a group of people diagnosed with any category of depression.
Minor issues:
Line 81. When it says “weeks of months” I guess it should say “weeks or months”.
Line 364. References (Zak, 2022; Hsu, 2012; Ferring et al., 2004) are missing in the list of references.
Lines 613-614. Reference 92 should include the year of publication.
Author Response
- Participants: Please, provide information about the recruitment procedure, as well as inclusion/exclusion criteria. The authors mention that the only demographic data collected was sex. Why other demographic information, particularly participants’ socioeconomic and health status was not collected?
The ms has been revised to include additional information about recruiting. In order to facilitate the burden of data collection, very little demographic or other information was collected from participants. The ms. states that health status was collected daily (Methods--Procedure).
- Please, provide information about how the current study accomplished the Declaration of Helsinki, domestic legislation for human research and the university ethics committee approval.
Added.
- Please provide information about participants abandonment and failure rates to complete the online survey.
The amount of missing data for the dependent variables is now included in the ms. and we note that all participants who started the study completed it.
- The survey items employed to record participants mood, health and energy should be provided or indicate whether it has been published elsewhere. Has such questionnaire been validated? Psychometric properties of the questionnaire should be provided.
The mood survey now includes a cite in Methods.
- Procedure: Although some references to previous articles are provided, a more detailed description about the construct “Immersion” and its physiological bases should be given. Furthermore, some references are missing in the References section (line 117: Zak & Barraza, 2018) and (line 119: Zak & Nowack, 2021; Zak et al. 2021).
The methods section has been expanded to discuss Immersion and references added.
- When (daytime) were the physiological measures registered?
The revised ms. notes that data are sent to the cloud continuously.
- What is the psychometric soundness of the items included in the questionnaire?
Addressed in #4.
- How the questionnaire correlates with a diagnostic of depression?
The original ms. (36-37) states that mood troughs lasting 2 weeks or more are considered as an indicator of depression. The revised ms. now states our goal was not to measure depression which is a state people stay in for periods of time but to classify low moods as a depression prodrome.
- Discussion: The authors acknowledge that a diagnostic of depression may take months as result of the variability of mood. I understand their model is proposed as a tool to help in the detection and subsequent treatment of depression in elders. Nevertheless, they have used a sample of healthy seniors. As a limitation of the study it should include a group of people diagnosed with any category of depression.
We have added this limitation to the Discussion.
10 Minor issues:
Line 81. When it says “weeks of months” I guess it should say “weeks or months”.
Line 364. References (Zak, 2022; Hsu, 2012; Ferring et al., 2004) are missing in the list of references.
Lines 613-614. Reference 92 should include the year of publication.
Thank you, these have been corrected.
Round 2
Reviewer 1 Report
The quality of paper is improved.